# Identification of Potential Therapeutics of *Mentha* Essential Oil Content as Antibacterial MDR Agents against AcrAB-TolC Multidrug Efflux Pump from *Escherichia coli*: An In Silico Exploration

**DOI:** 10.3390/life14050610

**Published:** 2024-05-09

**Authors:** Rawaf Alenazy

**Affiliations:** Department of Medical Laboratory, College of Applied Medical Sciences-Shaqra, Shaqra University, Shaqra 11961, Saudi Arabia; ralenazy@su.edu.sa

**Keywords:** *Mentha*, essential oils, antimicrobial MDR, *E. coli* AcrB-TolC, in silico molecular docking

## Abstract

Multidrug-resistant bacterial pathogens, such as *E. coli*, represent a major human health threat. Due to the critical need to overcome this dilemma, since the drug efflux pump has a vital function in the evolution of antimicrobial resistance in bacteria, we have investigated the potential of *Mentha* essential oil major constituents (**1**–**19**) as antimicrobial agents via their ability to inhibit pathogenic DNA gyrase and, in addition, their potential inhibition of the *E. coli* AcrB-TolC efflux pump, a potential target to inhibit MDR pathogens. The ligand docking approach was conducted to analyze the binding interactions of *Mentha* EO constituents with the target receptors. The obtained results proved their antimicrobial activity through the inhibition of DNA gyrase (**1kzn**) with binding affinity ΔG values between −4.94 and −6.49 kcal/mol. Moreover, *Mentha* EO constituents demonstrated their activity against MDR *E. coli* by their ability to inhibit AcrB-TolC (**4dx7**) with ΔG values ranging between −4.69 and −6.39 kcal/mol. The antimicrobial and MDR activity of *Mentha* EOs was supported via hydrogen bonding and hydrophobic interactions with the key amino acid residues at the binding site of the active pocket of the targeted receptors.

## 1. Introduction

Worldwide, we are noticing an outstanding increase in bacterial resistance to a wide range of antibiotics due to the indiscriminate use of commercial antimicrobial agents. This forces our attention to search for new antibiotics and/or antibacterial agents to treat infectious diseases [1,2].

Significantly, through the 21st century, severe bacterial infections have become resistant to the frequently used antibiotics [3]. Most of the discovered antibiotics so far have become inefficient in overcoming bacterial resistance; new genes and transmission vectors of the bacteria that are identified on a regular basis are encoded antibiotic resistance [4].

Bacterial resistance to antibiotics is achieved via varied and complicated molecular mechanisms; the most common is horizontal gene transfer [5]. Besides that, new mechanisms of bacterial resistance have not been identified yet, which led to the term ‘superbugs’ for multidrug-resistant bacterial strains. The principal factors for such resistance come from misuse and/or overuse of antimicrobial agents [4].

Drug efflux pumps (EPs) have a vital function in the evolution of antimicrobial resistance (AMR) in bacteria [6]. Their ability to extrude antimicrobials, prevent the accumulation of toxic levels of antibiotics, and grant pathogen nonsusceptibility to antimicrobial sources among Gram-negative bacteria [7,8].

Clinically, AcrAB-TolC is the major resistance–nodulation–division (RND) efflux system in *E. coli*, *Salmonella,* and other members of Enterobacteriaceae [9]. AcrB can efflux out norfloxacin, ciprofloxacin (hydrophobic fluoroquinolones), tetracyclines, etc. [10]. Hence, the increased expression of RND homogeneous EPs, especially AcrAB-TolC, confers MDR, which is observed broadly in both human and animal pathogens [11].

This development in the antibiotic resistance gained by human pathogens has driven the search for new agrochemicals, antibacterial agents, and chemotherapeutics that may integrate higher antimicrobial efficacy, lower toxicity, and minimize the negative impact on the environment.

Natural products (NPs), the plant secondary metabolites, i.e., phenolics, terpenoids, essential oils, alkaloids, etc., have been used for centuries as the main source of medicine used to treat and cure all sorts of diseases. They serve as plant defense mechanisms against bacteria, herbivores, and insects, besides their integrative therapeutics with biological systems. Moreover, most of today’s marketed drugs are natural-based products or their derivatives [12].

Essential oils represent one large group of plant phytoconstituents that recorded potent antimicrobial activity against Gram-positive and Gram-negative bacteria. Moreover, it succeeded in inhibiting the growth of MDR bacterial strains, as witnessed by in vitro tests, especially in agar diffusion, agar or broth dilution, and vapor phase tests [13].

Briefly, essential oils destabilize the cellular architecture, inducing membrane integrity breakdown and then increasing permeability, which results in the disruption of many metabolic regulatory functions such as cellular activities, including energy production (membrane-coupled) and membrane transport. This disruption affects nutrient processing, the secretion of growth regulators, the synthesis of structural macromolecules, and various vital processes [14]. Moreover, owing to the lipophilic and hydrophilic nature of essential oils, which enable their assimilation into the cell membrane, they may affect the external envelope of both the cell and cytoplasm as they penetrate the bacterial cell membrane, resulting in the cell organelles being affected [15].

The genus *Mentha* belongs to the family Lamiaceae, and it consists of around 25 species distributed in Africa, Asia, Australia, Europe, and North America [16], from which *M. piperita* L., *M. spicata* L., *M. lavendulacea* Willd, *M. longifolia* L., and *M. microphylla* C. Koch are widely distributed in KSA [17,18]. *Mentha* spp. has a long history of folkloric uses to treat gastrointestinal tract disorders, alleviate chest and stomachache pains, stimulate digestion, treat aerophagia, biliary disorders, dyspepsia, enteritis, flatulence, gallbladder spasms, gastric acidities and gastritis, intestinal colic, spasms of the bile duct, and treat obesity [19,20,21].

Numerous studies have clearly revealed the antibacterial and antifungal activities of *Mentha* species [22] and concluded that their essential oils are more efficient antifungals and antibacterial agents compared to other polar extracts [23,24,25].

*Mentha* spp. essential oil is widely used as an antimicrobial, an additive to analgesic creams, and to treat oral mucosal inflammation. Moreover, it is also used to treat bile duct discomfort and menstrual cramps, expectorant, irritable bowel syndrome, myalgia and neuralgia, secondary amenorrhea, and oligomenorrhea [26,27].

*Mentha* spp. essential oils are distributed over the leaves, stems, and their reproductive structures, and their composition differs widely depending on their origin, geographic regions, and planting environment [24]. Anyhow, they are rich in oxygenated compounds with either C-2 (i.e., carvone and related compounds) or C-3 (i.e., menthone, piperitone, piperitenone, and pulegone), which are characteristic of a free hydroxyl group and possess a system of delocalized electrons that account for their antimicrobial activity as they act as a proton exchanger, reducing the pH gradient across the cytoplasmic membrane and resulting in destabilizing the cytoplasmic membrane. Moreover, it led to the collapse of the proton motive force and the depletion of the ATP pool, which eventually led to cell death [28].

According to the above-mentioned data, our aims were to prove the reported in vitro antimicrobial potential of *Mentha* EOs via in silico molecular docking of EOs constituents and DNA gyrase, a potential protein target for the pathogen’s transcription and replication. In addition, an in silico exploration of the EO main constituents against the *E. coli* AcrB-TolC efflux pump was conducted to demonstrate the antibacterial activity of *Mentha* spp. essential oil against the MDR *E. coli* AcrB-TolC efflux pump.

## 2. Materials and Methods

### 2.1. In Silico Molecular Docking Study

#### 2.1.1. Bioinformatics Tools

Open Bable GUI (Open Babel—the chemistry toolbox—Open Babel openbabel-3-1-1 documentation (https://openbabel.org/)), Discovery Studio Client (v2021; A Product of Accelrys Inc., San Diego, CA, USA), PerkinElmer ChemOffice Suite 2020 v20.1.1.125, and AutoDock 4.2.6 software (AutoDock (https://autodock.scripps.edu/)).

#### 2.1.2. Protein Preparation

The X-ray crystallographic structure of protein targets (PDB ID: **4dx7** (responsible for transport of drugs by the multidrug transporter AcrB involves an access and a deep binding pocket that are separated by a switch-loop in complex with doxorubicin) and **1kzn** (Crystal structure of *E. coli* 24 kDa domain in complex with clorobiocin) were downloaded from the RCSB protein data bank (https://www.rcsb.org/). Non-essential molecules of water, heteroatoms, and co-crystallized ligands bound to the receptors were deleted. Moreover, all hydrogen and missing atoms were added to the receptor molecule’s target. Subsequently, Kollman united atom charges were assigned to the receptor atoms [29]. Binding pockets with the key amino acids of the selected target proteins were predicted based on their co-crystallized, pounded ligands. Grid boxes were built around the binding sites manually for **4dx7** and **1kzn** (Centre: X: 27.974 and 18.411, Y: −37.688 and 25.268, Z: −9.594 and 37.049 Å, respectively, and dimensions: x: 60, y: 60, z: 60), with a grid spacing of 0.5 Å. These dimensions covered the whole binding site and provided enough space for ligand translation and rotation. The corresponding grid center coordinates were set according to the respective binding site residues of the proteins.

#### 2.1.3. Ligands Preparation

The 3D structures of the main essential oil constituents of *Mentha* spp. (Table 1) were retrieved in (.sdf) format from PubChem (www.pubchem.com), then converted into their (.pdbqt) files using Open Babel software, which is freeware. Subsequently, Gasteiger charges were added to each atom, and the maximum number of rotatable bonds was set according to the torsional bonds in each compound.

#### 2.1.4. Docking of the Receptors with the Ligands

The virtual docking of the selected ligands against target proteins was evaluated by AutoDock 4.2.6 software [29]. Firstly, a re-docking process of the original co-crystallized ligand(s), i.e., doxorubicin and clorobiocin of **4dx7** and **1kzn**, respectively, was performed for docking validation, which is well reproduced with RMSD values of 0.00 Å and binding energy values of −8.42 and −6.73 kcal mol^−1^, respectively. The docking study was performed using the Lamarckian genetic algorithm, with 50 as the total number of GA runs. In each respective run, a population size of 300 individuals with 27,000 generations and 2,500,000 energy evaluations was employed. Operator weights for crossover, mutation, and elitism were set to 0.8, 0.02, and 1, respectively. The single docked conformation was selected from each docking round based on the clustering RMSD (≤2 Å) and lowest binding energy. The most stable conformations of the ligand molecule were selected based on the lowest binding energy and their binding mode at the active site of proteins, and the 2D and 3D binding interactions of the (.pdbqt) complexes of protein–ligand were analyzed using Discovery Studio Client (Discovery Studio Client is a product of Accelrys Inc., San Diego, CA, USA).

## 3. Results and Discussion

### 3.1. Antimicrobial Exploration of Mentha Essential Oil

Compounds that characterize the different *Mentha* chemotypes are those commonly occurring as main components of *Mentha* essential oil (Table 1); their formation reflects differences in biosynthetic pathways, together with a few other compounds that have also been reported sporadically as main *Mentha* spp. oil components. These variations, of course, accounted for their antibacterial potential with respect to one pathogenic bacteria species; it is well known that EO is not a single compound but a combination of the chemical compounds that carry the specific antimicrobial activity [30,31].

One of the major distinctivenesses of essential oils is their hydrophobicity, which facilitates their penetration into the cell membrane and influences the external envelope of both the cell and cytoplasm, resulting in the cell organelles being affected [15]. *Mentha* EO constituents (Table 1) are characterized by the presence of a free hydroxyl group (oxygenated monoterpenes), which helps in the formation of delocalized electrons, resulting in a proton exchanger, reducing the pH gradient, and destabilizing the cytoplasmic membrane; hence, breakdown of the proton motive force and then drop of the ATP pool, which eventually leads to cell death [28].

*Mentha* spp. essential oils showed remarkable antimicrobial potential against bacteria, fungi, and other microorganisms, such as yeasts and periodontopathogens [26], mainly due to the presence of oxygenated monoterpenes in their chemical compositions, with bactericidal and bacteriostatic concentration ranges of 1/1 to 1/1000 (*v*/*v*) and 1–5 mg/mL, respectively [32]. Briefly, EOs of *M. rotundifolia* exhibited strong antimicrobial effects against *B. subtilis*, *B. cereus*, *E. coli*, *P. mirabilis*, *S. typhimurium*, and *S. aureus* [33,34,35]. Moreover, *M. suaveolens* efficiently inhibited 20 strains of microorganisms [36]. Furthermore, the Algerian *M. pulegium* recorded antimicrobial potential against a wide number of Gram +ve, three Gram −ve, fungal strains, and yeasts [24,37].

Nevertheless, the EO of *M. piperita*, *M. pulegium*, and *M. spicata* proved appreciable activity against *C. albicans*, *E. coli*, *S. aureus*, *S. pyogenes* [38], and *S. pyogenes* [39]. Sixteen microorganisms, including *E. coli*, *Shigella sonnei*, *Micrococcus flavus*, etc., were inhibited by the EOs of *M. longifolia* [24], *M. aquatica*, and *M. piperita* [40], and *M. arvensis* [41]. In addition, *M. officinalis* EOs totally inhibit *E. coli*, *B. aureus*, *S. lactis*, and *S. aureus* [42]. Moreover, *Mentha* spp. EOs have been considered a safe ingredient for the development of antibiofilm agents that could find a role in the pharmaceutical industry [26].

The above-recorded findings about the potential antimicrobial activity of *Mentha* spp. EOs have gained our attention to study their mechanism of action in detail from the side of in silico molecular docking and prove their activity against *E. coli* MDR.

### 3.2. Molecular Docking Analysis

The crucial role in structure-based drug design (SBDD) is the molecular docking approach, which is used better to understand and estimate the molecular interactions, binding affinities, and energies of ligand(s) within a targeted protein [43]. It should be noted that antibiotics target microbial metabolism and restrict their growth by deactivating the vital enzymes involved in cell wall biosynthesis and repair.

Briefly, DNA gyrase is pivotal for bacterial survival as it controls DNA topology by introducing transient breaks to both DNA strands during transcription and replication, so it is essential to exploit bacterial DNA gyrase as a critical target for antibacterial agents. Additionally, bacterial strains develop MDR against antibiotics by effluxing it out through AcrB-TolC. Consequently, a molecular docking study was carried out to examine the binding interactions of the major volatile constituents with the key pockets of regulatory enzymes **4dx7** and **1kzn**.

In the present study, we selected a total of 19 essential oil constituents (Table 1, Figure 1) reported to be the main EO composition of *Mentha* spp. [21], which were picked up as powerful natural sources of antimicrobials.

The docking results of this study revealed the best-docked conformer of the ligands within the target receptor based on the predicted binding affinity, hydrogen bond, and hydrophobic interactions. Briefly, as reported by Aouf and co-workers [44], the key amino acids of the active pocket of DNA gyrase (PDB ID: **1kzn**) conserved Val43, Asp46, Val47, Val71, Asp73, Ile78, Pro79, Ala90, Ala94, Met95, Val120, Thr165, and Val167 at the binding site. Hence, by reviewing molecular interactions, we can clarify that all ligands lie inside the binding pocket of DNA gyrase (**1kzn**), considering one of their respective conformers’ most energetically favorable binding interactions and exhibiting plausible binding having ΔG values between −4.94 and −6.49 kcal/mol, as shown in Table 1, from which we obtain two ligands, namely neo-iso-menthol and linalyl acetate, which revealed two H-bond interactions with Asp73 (1.65 Å) and Thr165 (3.08 Å) and Gly77 (2.14 Å) and Thr165 (2.00 Å), respectively.

Moreover, the key amino acids Val43, Val71, Thr165, and Val167 at the binding site of the active pocket formed one H-bond interaction with six ligands, namely carvone (Val167, 2.48 Å), linalool (Val43, 1.94 Å), menthol (Val43, 2.06 Å), menthyl acetate (Thr165, 2.05 Å), piperitoneoxide (Thr165, 2.95 Å), and P-menth-2-en-ol (Val71, 2.08 Å), besides some other hydrophobic interactions. Furthermore, the rest of the list (Table 1) displayed a number of hydrophobic interactions ranging from 4 to 7 (alkyl) with the key amino acids at the active site. Piperitoneoxide (HB, Thr165, 2.95 Å), menthofuran (C–HB, Val71, 2.89 Å), and carvone (HB, Val167, 2.48 Å) (Figure 2) were depicted as the top three binders with the highest molecular interactions (H-bond, C–HB, π-σ, π-alkyl, and/or alkyl) with 8, 7, and 7 interactions, respectively, with binding affinities of −5.17, −5.77, and −5.57.

On the other hand, as reported by Phan and others [45], the 3D structure of *E. coli* AcrB-TolC (**4DX7**) was selected as the target protein to unveil the inhibitory potential of *Mentha* EOs against *E. coli* AcrB-TolC MDR. This selection was supported by a previous recommendation by Abdel-Halim and colleagues [46], who examined the sequence alignment of all reported AcrB sequences against the *E. coli* AcrB sequence and confirmed a high similarity in the overall structure with a high conservation of the residues at the binding site, which is the same in **4DX7**. Additionally, **4DX7** was co-crystallized with doxorubicin. Nevertheless, all previous reports supported that the active pocket’s binding site of AcrB (Figure 3a) is large and encompasses mainly hydrophobic amino acids (i.e., Ala, Gly, Leu, Ile, and Phe), with some ionized and polar residues (i.e., Gln, Ser, Tyr, and Thr).

In addition, Nikaido [47] confirmed that this diversity is essential for the binding of various AcrB substrates and/or inhibitors, which should be characterized with hydrophobic groups to form hydrophobic interactions and atoms to create hydrogen bonds (Figure 3b). 

Briefly, ligand molecules depicted molecular interactions with the key amino acids at the fragment binding site of the active pocket, with ΔG values in the range of −4.69 to −6.39 kcal/mol. 

The most important amino acid residues that rendered stability to the bound ligand and protein complexes and were common in almost all ligands’ interactions are Tyr182, Tyr275, Ile277, Ile278, Leu293, and Phe617 that formed a hydrophobic trap, with which the frontrunner compounds (carvone, iso-pulegone, limonene, menthofuran, menthol, menthyl acetate, piperitenone, piperitenoneoxide, piperitone, *P*-menth-2-en-ol, pulegone) recorded alkyl, and/or π-alkyl interactions. Meanwhile, Table 1 revealed the predominant H-bond interactions, briefly, one HB for cineol (Gly296, 2.01 Å), iso-menthone (Gly272, 1.84 Å), iso pulegone (Gln151, 2.04 Å), linalyl acetate (Thr624, 1.96 Å), menthone (Ile277, 2.55 Å), piperitenone (Gln151, 2.04 Å), piperitone (Gln151, 1.94 Å), piperitoneoxide (Gly272, 2.50 Å) and *P*-menth-2-en-ol (Ile277, 2.07 Å), two HBs for carvone (Ser155, 2.59 Å; Ser180, 1.71 Å), linalool (Gln151, 1.74 Å; Ile277, 1.69 Å), menthofuran (Gln151, 2.13 Å; Ser155, 2.48 Å), menthol (Gln151, 1.83 Å; Ile277 2.24 Å), pulegone (Gln151, 2.16 Å; Ser180, 2.51 Å) and neoisomenthol (Gln151, 1.87 and 2.08 Å). Moreover, three HBs were recorded for menthyl acetate (Gln151, 2.32 Å; Ser155, 2.30 Å; Ser180, 1.92 Å) and piperitenoneoxide (Gln151, 1.83 Å; Ser155, 2.66 Å; Ser180, 2.37 Å). Hence, and from the frontrunner list, menthyl acetate, menthofuran, and piperitenoneoxide (Figure 4) were portrayed as the top three best binders, having free energy of binding of −6.07, −6.04 and −5.94 kcal/mol and molecular interactions (H-bond, π-alkyl) of 6, 6, and 8, respectively. Additionally, Gln151, Ser155, and Ser180 represented the best key amino acids in the active pocket for HB interactions, and Tyr275, Tyr182, and Ile278 represented the best key amino acids for alkyl interactions, as shown in Figure 4.

## 4. Conclusions

It can be concluded from the current study that *Mentha* essential oil constituents possess potential antimicrobial activity, which is well documented via wet lab measurements and confirmed by in silico molecular docking of the EO constituents into the active site of DNA gyrase. DNA gyrase is a key to pathogenic DNA topology during transcription and replication. Moreover, the antibacterial MDR activity of *Mentha* EOs was also evaluated here for the first time against *E. coli* AcrB-TolC and proved potential inhibition with proper binding affinities and varied molecular interactions at the active site of the hydrophobic pocket. The current study concludes that *Mentha* spp. EOs (**1**–**19**) exert their antimicrobial activity via their ability to inhibit the DNA gyrase of the pathogens and, consequently, interrupt its transcription and replication. Furthermore, *Mentha* spp. EOs proved their ability to restrain the MDR pathogens through their ability to suppress the AcrAB-TolC efflux pump. Hence, EOs (**1**–**19**) work synergistically to interrupt the transcription and replication of microbial pathogens and disturb their drug efflux pump.

## Figures and Tables

**Figure 1 life-14-00610-f001:**
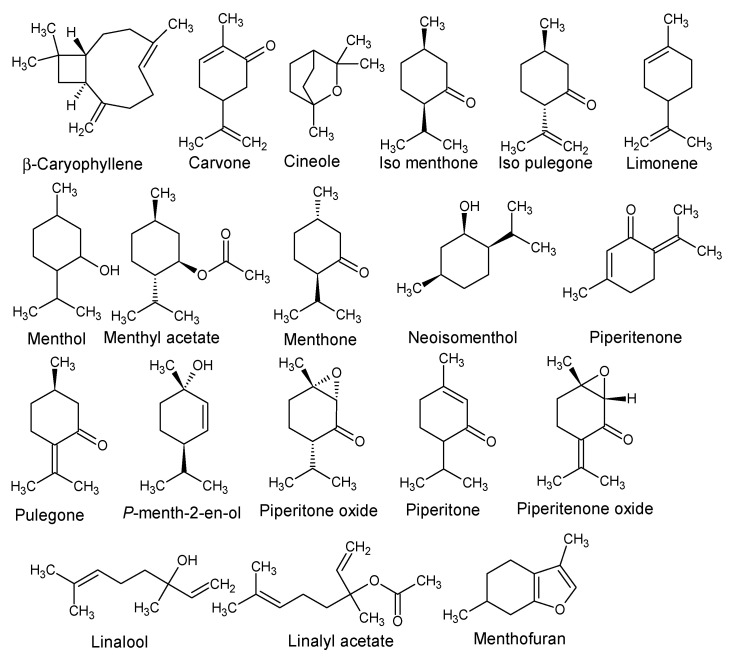
Chemical structures of the major EOs of *Mentha* spp.

**Figure 2 life-14-00610-f002:**
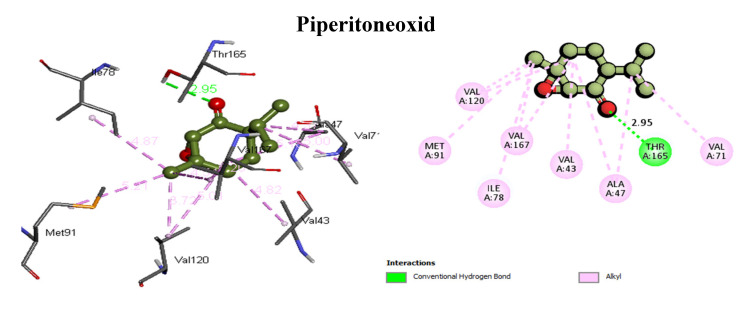
Three-dimensional and two-dimensional molecular interactions of piperitoneoxide, menthofuran, and carvone with the key amino acids at the active pocket of DNA gyrase (PDB: **1kzn**).

**Figure 3 life-14-00610-f003:**
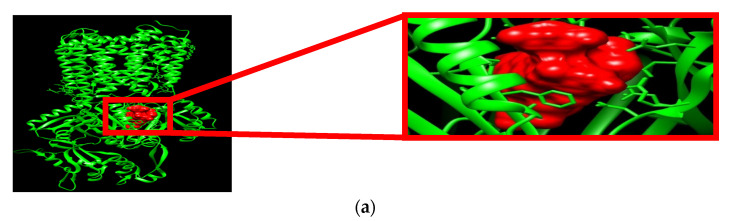
(**a**) **4DX7**_Chain-A and its hydrophobic active pocket (in red). (**b**) Hydrophobic active pocket with the degree of hydrophobicity.

**Figure 4 life-14-00610-f004:**
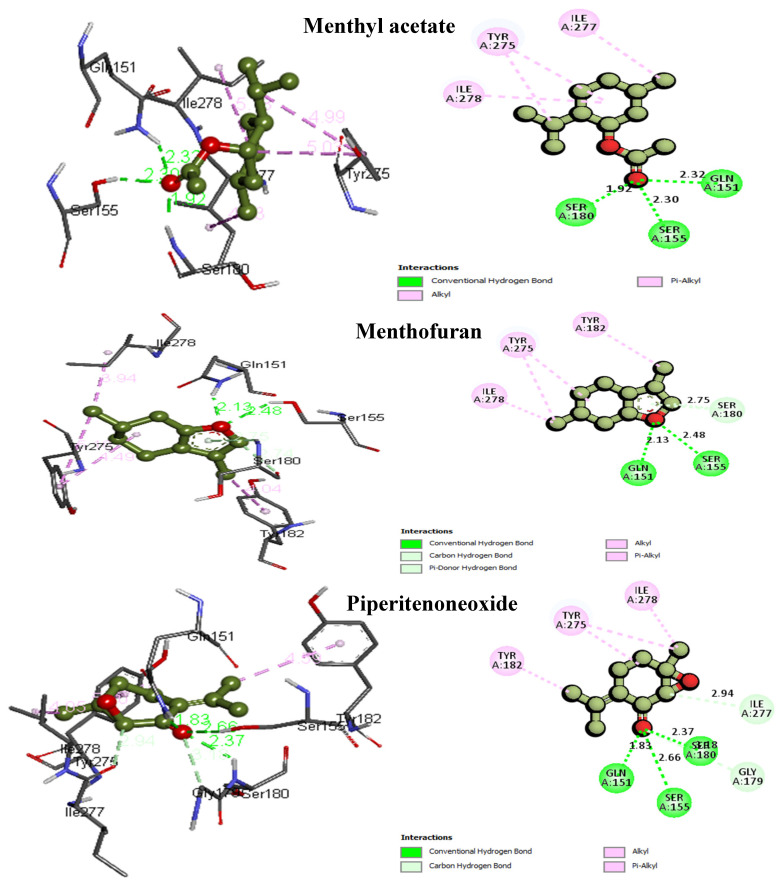
Three-dimensional and two-dimensional molecular interactions of menthyl acetate, menthofuran, and piperitenone oxide with the key amino acids at the active pocket of AcrB-TolC (PDB: **4dx7**).

**Table 1 life-14-00610-t001:** Binding free energies, hydrogen bonds, and number of interactions between the amino acid residues and the docked molecules into the binding sites of **4DX7** and **1KZN**.

Ligands	4DX7				1KZN			
	∆G *^1^	HBs	Molecular Interactions between the Amino Acid Residues and the Docked Ligands at the Binding Site		∆G *^1^	HBs	Molecular Interactions between the Amino Acid Residues and the Docked Ligands at the Binding Site	
			No. and Type of Interactions	Interacted Amino Acids			No. and Type of Interactions	Interacted Amino Acids
Doxorubicin *^2^	−8.42	7	13 (7 HB, 2 CHB, 1 π-HB, 1 π-σ, 1 π-anion, 1 π-alkyl)	Thr87, Gln89, Glu130, Lys163, Gln176, Asn274, Arg620 (HB) Asp174, Gln176 (C–HB) Gln176 (Pi-DHB) Leu177 (Pi-Sigma) Glu273 (Pi-Anion) Phe615 (Pi-Alkyl)				
Clorobiocin *^2^					−6.73	2	13 (1 vdW, 2 HB, 1 π-anion, 1 π-HB, 1 Amide-π, 7 π-alkyl)	Ala47 (van de Waals) Asp73, Gly77, Gly177 (HB) Glu50 (Pi-Anion) Thr165 (Pi-DHB) Asn46 (Amide-Pi Stacked) Val43, Ile78, Ile90, Ala96, Val118, Val120, Val167 (Alkyl and Pi-Alkyl)
1-*β*-Caryophyllene	−5.96	0	4 (4 Alkyl)	Ala580, Leu721, Pro814, Arg815 (Alkyl)	−6.07	0	6 (6 Alkyl)	Val43, Ala47, Ile78, Ile90, Val120, Val167 (Alkyl)
2-Carvone	−6.39	2	6 (2 HB, 1 CHB, 3 π-alkyl)	Ser155 (2.59 Å), Ser180 (1.71 Å) (HB) Ser155 (2.70 Å) (C–HB) Tyr182, Tyr275, Ile278 (Alkyl and Pi-Alkyl)	−5.57	1	7 (1 HB, 6 Alkyl)	Val167 (2.48 Å) (HB) Ala47, Val71, Ile78, Met91, Val120, Val167 (Alkyl)
3-Cineole	−5.28	1	3 (1 HB, 2 Alkyl)	Gly296 (2.01 Å) (HB) Ala39, Leu293 (Alkyl)	−5.33	0	6 (6 Alkyl)	Val43, Ala47, Val71, Ile78, Val120, Val167 (Alkyl)
4-Iso-menthone	−6.07	1	3 (1 HB, 2 Alkyl)	Gly272 (1.84 Å) (HB) Tyr275, Ile278 (Alkyl)	−5.57	0	7 (7 Alkyl)	Val43, Ala47, Val71, Ile78, Met91, Val120, Val167 (Alkyl)
5-Iso-pulegone	−5.49	1	4 (1 HB, 3 π-alkyl)	Gln151 (2.04 Å) (HB) Tyr182, Tyr275, Ile278 (Alkyl and Pi-Alkyl)	−5.46	0	5 (5 Alkyl)	Val43, Ala47, Val71, Ile78, Val167 (Alkyl)
6-Limonene	−5.82	0	3 (3 π-alkyl)	Tyr182, Tyr275, Ile278 (Alkyl and Pi-Alkyl)	−5.41	0	5 (5 Alkyl)	Val43, Ala47, Val71, Ile78, Val167 (Alkyl)
7-Linalool	−4.76	2	5 (2 HB, 3 π-alkyl)	Gln151 (1.74 Å), Ile277 (1.69 Å) (HB) Tyr275, Ile277, Ile278 (Alkyl and Pi-Alkyl)	−4.94	1	6 (1 HB, 5 Alkyl)	Val43 (1.94 Å) (HB) Val43, Ala47, Val71, Ile78, Val167 (Alkyl)
8-Linalyl acetate	−4.69	1	3 (1 HB, 2 π-alkyl)	Thr624 (1.96 Å) (HB) Met575, Phe617 (Alkyl and Pi-Alkyl)	−5.59	2	6 (2 HB, 4 Alkyl)	Gly77 (2.14 Å), Thr165 (2.00 Å) (HB) Val43, Ala47, Val71, Val167 (Alkyl)
9-Menthofuran	−6.04	2	6 (2 HB, 1 π-HB, 3 π-alkyl)	Gln151 (2.13 Å), Ser155 (2.48 Å) (HB) Ser180 (Pi-H) Tyr182, Tyr275, Ile278 (Alkyl and Pi-Alkyl)	−5.77	0	7 (1 CHB, 1 π-σ, 5 π-alkyl)	Val71 (2.89 Å) (C–HB) Thr165 (Pi-Sigma) Val43, Ala47, Ile78, Val120, Val167 (Alkyl and Pi-Alkyl)
10-Menthol	−7.10	2	5 (2 HB, 3 π-alkyl)	Gln151 (1.83 Å), Ile277 (2.24 Å) (HB) Tyr182, Tyr275, Ile278 (Alkyl and Pi-Alkyl)	−5.75	1	5 (1 HB, 4 Alkyl)	Val43 (2.06 Å) (HB) Val43, Ala47, Val71, Ile78 (Alkyl)
11-Menthone	−6.37	1	4 (1 HB, 1 CHB, 2 π-alkyl)	Ile277 (2.55 Å) (HB) Ser180 (3.58 Å) (C–HB) Tyr275, Ile278 (Alkyl and Pi-Alkyl)	−5.74	0	4 (4 Alkyl)	Val43, Ala47, Val71, Ile78 (Alkyl)
12-Menthyl acetate	−6.07	3	6 (3 HB, 3 π-alkyl)	Gln151 (2.32 Å), Ser155 (2.30 Å), Ser180 (1.92 Å) (HB) Tyr275, Ile277, Ile278 (Alkyl and Pi-Alkyl)	−6.49	1	6 (1 HB, 5 Alkyl)	Thr165 (2.05 Å) (HB) Val43, Ala47, Val71, Val120, Val167 (Alkyl)
13-Neoisomenthol	−6.02	2	4 (2 HB, 2 π-alkyl)	Gln151 (1.87 and 2.08 Å) (2 HB) Tyr275, Ile278 (Alkyl and Pi-Alkyl)	−5.37	2	5 (2 HB, 1 CHB, 2 Alkyl)	Asp73 (1.65 Å), Thr165 (3.08 Å) (HB) Thr165 (3.07 Å) (C–HB) Ala47, Ile78 (Alkyl)
14-Piperitenone	−6.14	1	4 (1 HB, 3 π-alkyl)	Gln151 (2.04 Å) (HB) Tyr182, Tyr275, Ile278 (Alkyl and Pi-Alkyl)	−5.71	0	6 (6 Alkyl)	Ala47, Val71, Ile78, Met91, Val120, Val167 (Alkyl)
15-Piperitenoneoxide	−5.94	3	8 (3 HB, 2 CHB, 3 π-alkyl)	Gln151 (1.83 Å), Ser155 (2.66 Å), Ser180 (2.37 Å) (HB) Gly179 (3.18 Å), Ile277 (2.94 Å) (C–HB) Tyr182, Tyr275, Ile278 (Alkyl and Pi-Alkyl)	−5.69	0	4 (4 Alkyl)	Val43, Ala47, Val71, Val167 (Alkyl)
16-Piperitone	−6.26	1	5 (1 HB, 1 CHB, 3 π-alkyl)	Gln151 (1.94 Å) (HB) Gly179 (2.87 Å) (C–HB) Tyr182, Tyr275, Ile278 (Alkyl and Pi-Alkyl)	−5.55	0	7 (7 Alkyl)	Val43, Ala47, Val71, Ile78, Met91, Val120, Val167 (Alkyl)
17-Piperitoneoxide	−5.55	1	4 (1 HB, 1 π-lone pair, 2 Alkyl)	Gly272 (2.50 Å) (HB) Tyr275 (Pi-Lone Pair) Ile278 2(Alkyl)	−5.17	1	8 (1 HB, 7 Alkyl)	Thr165 (2.95 Å) (HB) Val43, Ala47, Val71, Ile78, Met91, Val120, Val167 (Alkyl)
18-*P*-menth-2-en-ol	−5.57	1	4 (1 HB, 3 π-alkyl)	Ile277 (2.07 Å) (HB) Tyr182, Tyr275, Ile278 (Alkyl and Pi-Alkyl)	−5.58	1	5 (1 HB, 4 Alkyl)	Val71 (2.08 Å) (HB) Val43, Ala47, Val71, Val167 (Alkyl)
19-Pulegone	−6.26	2	5 (2 HB, 3 π-alkyl)	Gln151 (2.16 Å), Ser180 (2.51 Å) (HB) Tyr182, Tyr275, Ile278 (Alkyl and Pi-Alkyl)	−5.77	0	6 (6 Alkyl)	Ala47, Val71, Ile78, Met91, Val120, Val167 (Alkyl)

*^1^: Binding affinities; *^2^ Co-crystallized ligands.

## Data Availability

Data is contained within the article.

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
