# Peer review of "Identification of Potential Therapeutics of Mentha Essential Oil Content as Antibacterial MDR Agents against AcrAB-TolC Multidrug Efflux Pump from Escherichia coli: An In Silico Exploration"

_life, 2024, doi:10.3390/life14050610_

Round 1
Reviewer 1 Report
Comments and Suggestions for Authors
1) The Article was well written and the results were discussed clearly.
2) It’s advised to change the background of the images from black to white
3) What about the hydrogen bond distances? Measure the distance of the hydrogen bond and include the distances in the table as well as in the results and discussion part.
4) Another suggestion is to run molecular dynamics simulation for at least 100ns.
5) There are no sufficient writings for the figures inside the content. It would be better if few more lines were written describing the figures.
6) Its also advised to include the interaction images with the hydrogen bond distance measurement.
Author Response
I have submitted my comments to the first reviewer here (Reviewer comments -1), please ignore the other files except this file

Reviewer 2 Report
Comments and Suggestions for Authors
Manuscript Number: life-2983631
entitled: Identification of Potential Therapeutics of Mentha Essential Oil Content as Antibacterial MDR Agents against AcrAB-TolC Multidrug Efflux Pump from Escherichia coli: An In-Silico Exploration
I had the great pleasure of reading the peer-reviewed article. The scientific research was well conducted, and the results were described accurately and concisely. Therefore, the manuscript is suitable for publication in Life; however, it needs some corrections:
1. Please add a chapter on abbreviations and explain all terms.
2. Please add the following information about the compositions of the main components of Mentha essential oil. The place where the essential oil is produced and where plants are growing affects the composition, latitude, or type of soil and sunlight.
3. Please add the chemical structures of the main components, maybe in Table 1.
Author Response
I have submitted my comments to the second reviewer here (Reviewer comments -2)

Reviewer 3 Report
Comments and Suggestions for Authors
Interesting and extensive work on the potential antimicrobial activity of the mentha essential oil and its mechanisms of action against MDR E. coli.
The objectives of the work should be more clearly identified both in the abstract and at the end of the introduction.
The literature review is extensive and relatively up-to-date although not systematic.
In the materials and methods section, please describe the used Bioinformatics tools in more detail.
The "Conclusions" section seems to be missing a final concluding sentence.
Minor comments:
-line 281: please replace "transcriptional" with "transcription"
Author Response
I have submitted my comments to the third reviewer here (Reviewer comments -3).

Round 2
Reviewer 1 Report
Comments and Suggestions for Authors
Accept in present form Manuscript